# Heterogeneous Deformation Behavior of Cu-Ni-Si Alloy by Micro-Size Compression Testing

**Sari Yanagida [1,2], Takashi Nagoshi [3,\*], Akiyoshi Araki [2], Tso-Fu Mark Chang [1,2], Chun-Yi Chen [1,2], Equo Kobayashi [2], Akira Umise [1,2], Hideki Hosoda [1,2], Tatsuo Sato [1,2] and Masato Sone [1,2,\*]**

[1] Institute of Innovative Research, Tokyo Institute of Technology, Kanagawa 226-8503, Japan; yanagida.sari@kobelco.com (S.Y.); chang.m.aa@m.titech.ac.jp (T.-F.M.C.); chen.c.ac@m.titech.ac.jp (C.-Y.C.); umise.a.aa@m.titech.ac.jp (A.U.); hosoda.h.aa@m.titech.ac.jp (H.H.); sato.tatsuo8@gmail.com (T.S.)

[2] Department of Material Science and Engineering, Tokyo Institute of Technology, Kanagawa 226-8503, Japan; a.akiyoshi.1111@gmail.com (A.A.); kobayashi.e.ad@m.titech.ac.jp (E.K.)

[3] National Institute of Advanced Industrial Science and Technology, Ibaraki 305-8564, Japan

\* Correspondence: nagoshi-t@aist.go.jp (T.N.); sone.m.aa@m.titech.ac.jp (M.S.); Tel.: +81-45-924-5043 (M.S.)

**Abstract:** The aim of this study is to investigate a characteristic deformation behavior of a precipitation strengthening-type Cu-Ni-Si alloy (Cu-2.4Ni-0.51Si-9.3Zn-0.15Sn-0.13Mg) by microcompression specimens. Three micropillars with a square cross-section of $20 \times 20 \times 40$ $\mu m^3$ were fabricated by focused ion beam (FIB) micromachining apparatus and tested by a machine specially designed for microsized specimens. The three pillars were deformed complicatedly and showed different yield strengths depending on the crystal orientation. The micromechanical tests revealed work hardening by the precipitation clearly. Electron backscattered diffraction analysis of a deformed specimen showed a gradual rotation of grain axis at the grain boundaries after the compression test.

**Keywords:** alloy; microcompression test; precipitation strengthening; work hardening; deformation

---

## 1. Introduction

When size of materials decreases from bulk size to micrometer scale, the dislocations could easily escape to the free surface before having interaction with each other, and the dislocation density would decrease, which leads to a dislocation starved condition and causes increased strength. This is a widely accepted explanation of size-dependent deformation behavior known as the "size effect" observed in crystalline metallic materials [1]. The investigation expands to variety of metallic materials such as noncubic metals [2,3] and nanocrystalline materials [4,5]. The dislocation motion could be affected by several internal defects such as vacancies [6], grain boundaries [5] and precipitations [7].

Uchic et al. [8] reported a microcompression testing method to effectively investigate mechanical property of small materials. Yet, in a compression test, the specimen is collapsed into a flat shape during the testing, and the rupture strength cannot be measured. Furthermore, buckling occurs easily and high aspect ratio specimens cannot be used. In this technique, the constraint between the specimen and indenter is only in one axis along the yield direction, so friction between the specimen and the indenter occurs and leads to heterogeneous stress distribution in the specimen [9]. To resolve this problem, Kiener et al. [10] developed a way to conduct microtensile testing. In this technique, the specimen's gripper part and the indenter are constrained in two directions [11], so homogeneity deformation of the specimen can be gained.

In this study, micromechanical properties of a precipitation-hardened copper alloy, Cu-Ni-Si alloy, are investigated. Cu-Ni-Si alloy is a precipitation strengthening-type alloy including δ-$Ni_2Si$ precipitates and receives a lot of attention for its applications for electronic components thanks to its

---

low cost, high strength and good electrical and thermal conductivity [12–14]. These properties are also beneficial for use in microelectromechanical systems (MEMS). However, effects of the precipitates in microscale are not yet thoroughly investigated. In our previous studies, we reported microtensile tests of a pure copper and a Cu-Ni-Si alloy with microtensile specimens fabricated by focused ion beam (FIB) milling system. In the microtensile tests, both of the specimens showed characteristic large serrations during deformation, which were not observed in the bulk samples, and the obvious necking deformation led to a decrease in the flow stress [15]. For deep understanding of micromechanical behavior in precipitate strengthened metals, comparisons of the results obtained from compression and tensile tests would be valuable to understand mechanical properties of the alloys in microscale in order to utilize them to the micro- or nano-electromechanical systems. In this study, the micromechanical property of Cu-2.4Ni-0.51Si-9.3Zn-0.15Sn-0.13Mg alloy under microcompression testing is discussed.

## 2. Materials and Methods

### 2.1. Sample Composition and Heat Treatment Process

The material investigated in this study was supplied by Furukawa Electric Inc. in a sheet form (1.2mm thickness) with the chemical composition Cu-2.4Ni-0.51Si-9.3Zn-0.15Sn-0.13Mg (mass%). For the Cu-Ni-Si alloy, a heat treatment process was conducted first. The heat treatment process involved homogenization at 1223 K for 18 ks, cold rolling for 90% in thickness reduction rate and solution treatment process at 1123 K for 600 s. In order to reveal the effect of the precipitation, aging at 723 K for 300 s was performed, where the temperature and time needed for the precipitation hardening were determined in previous studies [13,15]. The grain size of the Cu-Ni-Si alloy was about 30 μm.

### 2.2. Fabrication Process

The microsized specimens were fabricated by a FIB (FB2100, Hitachi, Tokyo, Japan) from the bulk sample described in former Section 2.1 [16,17]. In the beginning, the sliced sample with the thickness of about 50 m was fabricated from middle part of the bulk sample. For fabrication of the pillar, the specimen was tilted at −45 ± 2 degrees to avoid tapering of the pillar. For the compression test, three micropillars (specimens A, B and C) with different grain geometry as shown in Figure 1, having the identical size of 20 × 20 μm² in cross section and 40 μm in length were fabricated.

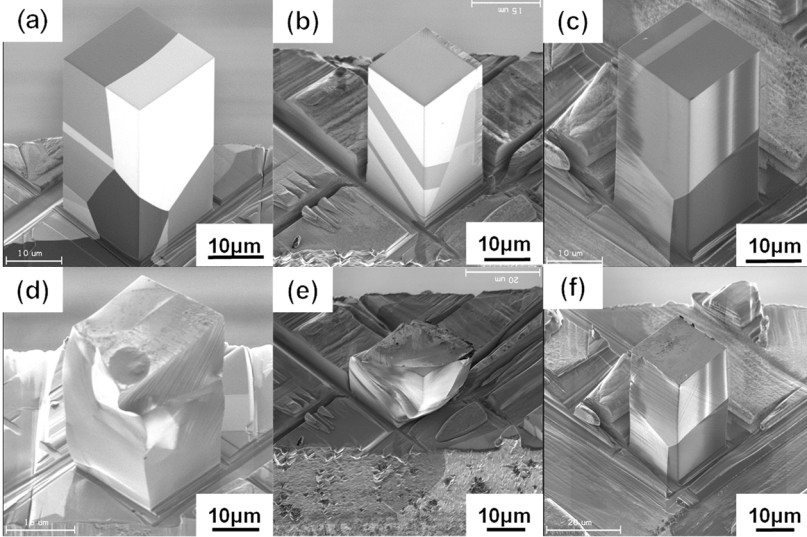

**Figure 1.** Scanning ion microscopy images of the three microcompression specimen AA: (**a**,**d**), specimen B (**b**,**e**), and specimen C: (**c**,**f**). The images were taken before: (**a**–**c**) and after; (**d**–**f**) the compression test.

*2.3. Micromechanical Testing*

A testing machine specially designed for microsized specimens was developed in our group [18]. The displacement was controlled at 0.1 µm/s by a piezoactuator. The test temperature was 295 ± 2K. The specimen was observed before and after the mechanical tests by a scanning electron microscope (SEM, S-4300 SE, Hitachi, Tokyo, Japan). After the test, a micropillar of the Cu-Ni-Si alloy was milled by the FIB to show cross-section of the deformed part and then observed by a scanning ion microscope (SIM) equipped in the FIB and analyzed by electron backscattered diffraction (EBSD, Bruker AXS GmbH, Karlsruhe, Germany and S-4300 SE, Hitachi, Tokyo, Japan).

## 3. Results and Discussion

*3.1. Mechanical Behavior by Microcompression Test*

SIM images of the three Cu-Ni-Si alloy micropillars before the compression test are shown in Figure 1a–c. SIM is the imaging technique using FIB in which crystallographic difference enhanced in their contrast thus can easily reveal grains and twin boundaries on surface of the micropillars. Figure 1d–f shows images of the three pillars after the microcompression test. The results indicated the deformation took place in all of the grains in the pillar. In SIM images of all of the deformed micropillars, thin, curvy and numerous slip lines with some small protrusion were observed. This tendency was also observed in pure Cu specimen which grains were constrained by grain boundaries [15].

Furthermore, after the deformation as shown in Figure 1d–f, the specimens were twisted a little. This shows grains in the microcompression specimen are constrained by grain boundaries causing the specimen to deform heterogeneously. Especially the black circle as shown in Figure 1d could be originated from the rotation deformation of a grain under microdiamond indenter. On the other hand, in a microtensile specimen of Cu-Ni-Si alloy as previous reported, the specimen was constrained in one direction only hence each grains had more freedom to deform, and the microtensile specimen deformed homogeneity [16].

Figure 2 shows the engineering stress-engineering strain ($S_E$-$S_E$) curves of the three specimens A, B and C. The yield strengths of specimens A, B and C were 278, 303 and 206 MPa, respectively. The difference in the yield stress was caused by dominantly crystal orientation difference, grain boundary, grain size and specimen size effect. In our previous work, the influence of orientations on mechanical properties of the microsized specimens of pure Cu was magnified with reduction of the number of grains inside a sample [15]. This size effect leads to scatter of experimental data of mechanical properties and can be quantified by the ratio between the specimen size: S and average grain size: G, denoted as S/G [19,20]. In this case, S/G ratio of single crystal specimen is "1". The deformation behavior of a single grain is strongly dependent on crystallographic orientation. On the other hand, the deformation of the bulk polycrystalline materials with large S/G ratio is isotropic because orientations of crystals are evenly and randomly distributed within the specimen. When the number of grains within a specimen becomes lower, it means with a S/G ratio smaller than 30, the effect of orientations of each grain and geometry of them becomes stronger in deformation behavior and the S/G ratio of the micropillars in this study was below 30. Thus the yield strength can be different in each pillar. In specimen C, the grains at the top of the pillar were relatively large and yielded in single grains thus having low yield strength and limited work hardening. The detailed effects of the crystal orientation and grain boundary on the grain deformation are discussed below.

The $S_E$-$S_E$ curves of the microcompression pillar also show work hardening during plastic deformation. Especially in the compression test, the pillar deformed into the shape of a barrel, which is another cause of the work hardening behavior in the engineering stress–strain behavior. In the microtensile study [15], typical serrations during the plastic deformation were observed in the $S_E$-$S_E$ curve. However, in the compression test, serrations were not observed and the curve is similar to the results obtained with a bulk specimen. The SEM image in Figure 1 showed that the microcompression specimen has smaller and more slip steps than that of the microtensile specimen.

Flat and heterogeneous deformation without any load drop in the microcompression specimen indicates that each grain deformed with small and many slip system activities. On the other hand, in the microtensile test, the deformation was homogeneous and the grains deformed with large slip, and this behavior led to large load drops during the plastic deformation.

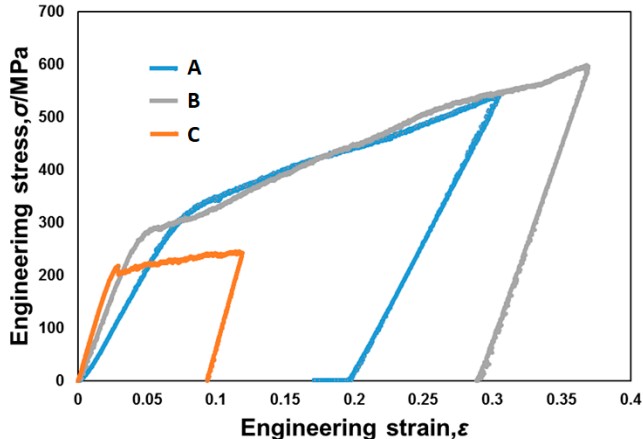

**Figure 2.** Engineering stress–engineering strain curves of microcompression tests for Cu-Ni-Si alloy specimens A, B and C.

### 3.2. Work Hardening Behavior of MicroCompression Test

Figure 3 shows work hardening behavior of specimen A in the microcompression test and indicates the enhanced work hardening. The work hardening rate was calculated by the following equation:

$$\Theta = \Delta\sigma/\Delta\varepsilon \tag{1}$$

using the true stress $\varepsilon$; and true strain $\sigma$; which were reduced from the engineering stress and the engineering strain observed, respectively. In the compression test, repetition of large bursts and drops could not be observed unlike the microtensile test reported in a previous study [15]. The steady state of work hardening rate curve observed at about 600 MPa may have been caused by the precipitation. The precipitates of δ-Ni$_2$Si forms as disc on (110) matrix plane and the orientation relationship between Ni$_2$Si and matrix is (110)Cu//(001) Ni2Si, [001]Cu//[010] Ni2Si [12,13]. They are reported to work as strong obstacles against the dislocation motion [12,13]. Nonshearable obstacles facilitate the dislocation multiplication by the Orowan mechanism, and generated dislocations are accumulated on the strong grain boundary which may lead to the enhanced work hardening in the Cu-Ni-Si alloy resulted in the steady state of work hardening rate curve.

Figure 4 shows three SIM images of specimen A taken from different directions. In the middle section of the edge part, complex distorted deformations with many shear lines were observed in Figure 4a. Figure 4b,c shows the side view of the distorted deformations, which are from the side plane of the pillar with a square cross section. These findings imply strain concentrates at the grain boundaries and/or near the grain boundaries, and the grain axis rotated gradually during the deformation. Figure 5 shows the crystal characteristic of the Cu-Ni-Si alloy specimen A after the compression test. The color in Figure 5b represents the orientations in inverse pole figure (IPF) against pillar axis shown in the inset. The orientations of grains α and β are plotted in IPF shown in Figure 5c and d. Grain α and β had near <110> and <111> orientations at surface, respectively. Both grains deformed heterogeneously and grain β showed gradual rotation with many slip lines in the side plane. This gradual rotation could come from the different compressive strength values of two or more grains with strong grain boundaries in the pillar. δ-Ni$_2$Si forms as disc on (110) matrix plane and the orientation relationship between Ni$_2$Si and matrix is (110)Cu//(001) Ni2Si, [001]Cu//[010] Ni2Si [12,13].

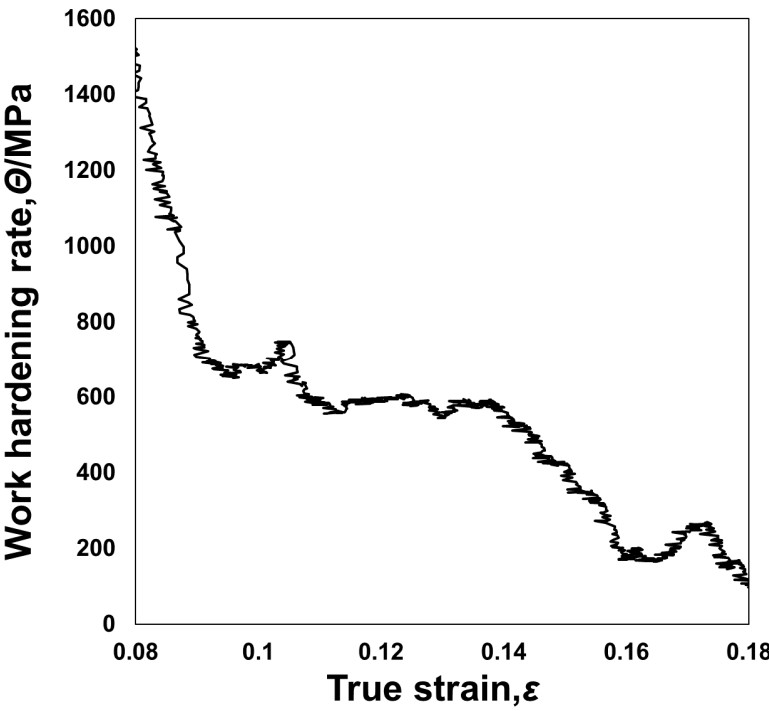

**Figure 3.** Work hardening behavior of Cu-Ni-Si alloy on microcompression test.

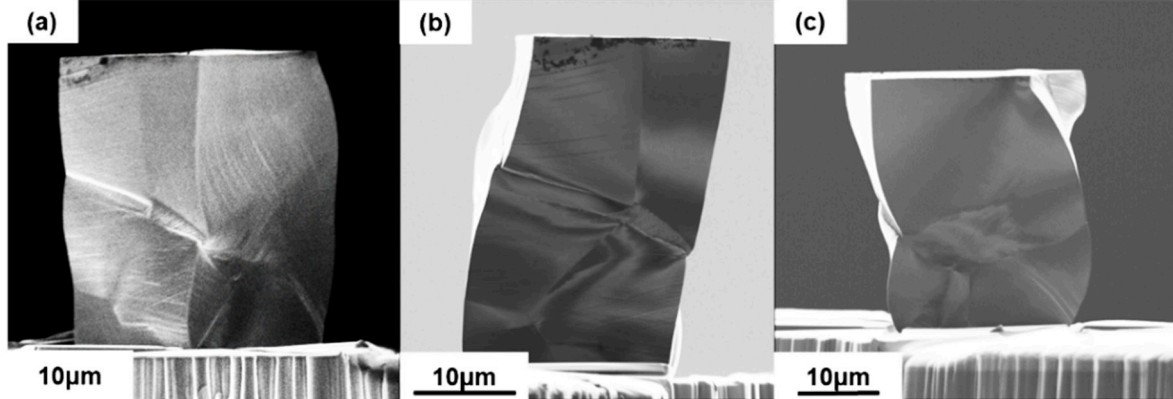

**Figure 4.** Scanning ion microscope (SIM) images of sample A in Cu-Ni-Si alloy specimens after microcompression test, observed taken from the front (**a**) and sides (**b**,**c**).

Figure 6 shows EBSD mapping of the longitudinal section (Figure 5a) of the deformed specimens. The change in color inside the grains shows gradual rotation of the crystal orientation and the orientation near grain boundaries not determined due to the distorted crystals by high density of dislocation at the boundary, which indicates high strength of the grain boundary in this alloy. In the compression test, stress concentrates at grain boundaries, and, near the grain boundaries, the grain axis rotated gradually. This result suggests that the grain boundary could be strengthened by the δ-Ni$_2$Si precipitates. The precipitates are reported to have formed in a shape of disk lying on {110} planes and are not shearable by dislocations, thus, they work as strong obstacles against the dislocation motion [12,13].

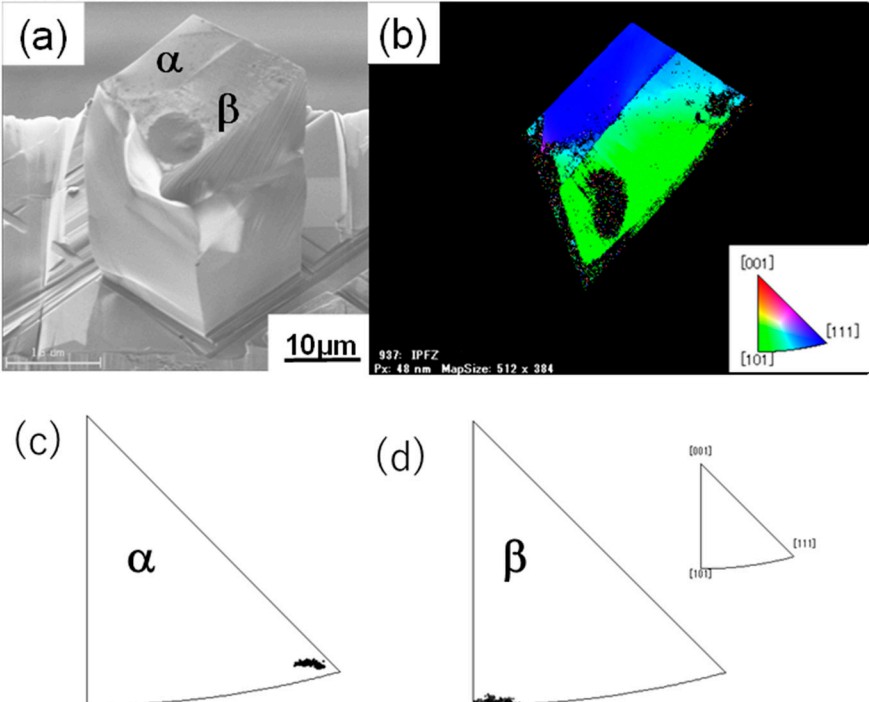

**Figure 5.** Crystal characteristic of the Cu-Ni-Si alloy specimen A after the compression test observed by (**a**) SIM and (**b**) electron backscattering diffraction (EBSD) image (ND) with inverse pole figures (Z axis) of grains α and β shown in (**c,d**) respectively.

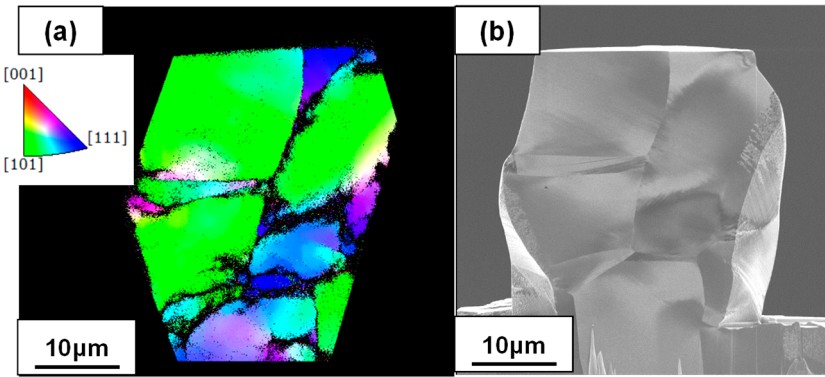

**Figure 6.** Crystal orientation analysis on longitudinal section of Cu-Ni-Si alloy specimen (correspond to Figure 5a) after the compression test observed by (**a**) EBSD mapping and (**b**) SIM image.

## 4. Conclusions

Microcompression tests of Cu-Ni-Si alloys were conducted. The mechanical tests of three micropillars showed different yield strength values ranged from 200 to 300 MPa, and this difference is considered to be contributed by the size effect. All of the work hardening curves exhibit plateau regimes, which is similar to the bulk specimen, and the specimen deformed heterogeneously. In the microcompression test, the SIM images showed small slip steps, which mean heterogeneous deformation by less restriction between the indenter and the specimen. EBSD analysis indicated a typical gradual rotation of crystal orientation and high accumulation of dislocation at the grain boundaries which imply enhanced dislocation multiplication at precipitates by the Orowan mechanism. The study showed precipitation effectively strengthened the metals and enhanced work hardening even in the microspecimens with few grains inside. The findings indicate the practical applications of precipitation-hardened materials for microcomponents used in MEMS.

**Author Contributions:** Conceptualization, S.Y., T.-F.M.C.; Methodology, S.Y., A.U., T.N.; Validation, C.-Y.C., E.K.; Formal analysis, S.Y.; Investigation, S.Y., T.-F.M.C.; Resources, A.A., T.S.; Data curation, S.Y., T.N.; Writing—original draft preparation, S.Y., Writing—review and editing, T.N., T.-F.M.C., M.S.; Visualization, S.Y., Supervision, T.-F.M.C., T.S., M.S.; Project administration, H.H., M.S.; Funding acquisition, T.S., H.H., M.S. All authors have read and agreed to the published version of the manuscript.

**Funding:** This research was funded by the Grant-in-Aid for Challenging Research (Pioneering) (JSPS KAKENHI Grant number 20K20544) and CREST Project (JST CREST Grant Number JPMJCR1433) by the Japan Science and Technology Agency. This work was performed under the support of the Science and Technology Agency (STA) of the Japanese government.

**Conflicts of Interest:** On behalf of all of the co-authors, the corresponding author states that there is no conflict of interest.

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
