# Peer review of "Heterogeneous Deformation Behavior of Cu-Ni-Si Alloy by Micro-Size Compression Testing"

_crystals, doi:10.3390/cryst10121162_

Round 1

Reviewer 1 Report

1)Precipitates were discussed in the manuscript, but no figures have been presented to support the claim. Precipitates can have complex influence on the deformation behaviro.  The effect of precipitates or second phase particles on deformation behavior and microstructure have actually been summarized in a review paper published at Prog Mater Sci:Progress in Materials Science 92(2018)284-359

2) The different yield strengths were ascribed to the different grain orientations within the micro-pillars. It is easy and important to also scan the initial grain orientations before the compression using EBSD. 

3) The effect of twin boundaries on the deformation behavior should be further elaborated.

4) Lines 179-181 are almost exactly the same as Lines 146-149.

Author Response

Answers to the comments by Reviewer #1

We would like to express special thanks for your questions and comments which are reasonable and important. On the all comments, we revised the text and reply the answer. We believe the revised manuscript could meet the all the comments by reviewers.

Comment #1: Precipitates were discussed in the manuscript, but no figures have been presented to support the claim. Precipitates can have complex influence on the deformation behavior.  The effect of precipitates or second phase particles on deformation behavior and microstructure have actually been summarized in a review paper published at Prog Mater Sci:Progress in Materials Science 92(2018)284-359

Response #1:

Thank you for the comment and the recommendation of a reference. This paper is cited as a reference in this manuscript.

Revision made:

Reference:

  1. Huang, K.; Marthinsen, K.; Zhao, Q.; Logéa, R.E. The double-edge effect of second-phase particles on the recrystallization behaviour and associated mechanical properties of metallic materials. 2018, 92, 284-359.

Comment #2:

The different yield strengths were ascribed to the different grain orientations within the micro-pillars. It is easy and important to also scan the initial grain orientations before the compression using EBSD. 

Response #2:

Orientation of the grains on pillar top can be easily evaluated by EBSD as shown in figure 5, however, evaluation of other grains not facing the pillar top is difficult due to pass of the electron beam diffraction. Yield strength of the single crystalline pillar is ascribed to the grain orientation, the pillars in this work has several grains and grain boundaries inside pillars, thus the yield strengths can not be easily assumed. The detailed description about yield strength of pillar added in the manuscript.

Comment #3:

The effect of twin boundaries on the deformation behavior should be further elaborated.

Response #3:

Twin boundaries are expected to have similar contribution to the deformation behavior as grain boundaries. The twin spacing observed in the micro-pillar is in the same level as the grain size, hence, the influence on the deformation behavior is expected as the grain size observed in the micro-pillar.

Comment #4:

Lines 179-181 are almost exactly the same as Lines 146-149.

Response #4:

Thank you for pointing out the mistake. Revision has been made.

Reviewer 2 Report

The work is very interesting and the results of EBSD deserve special attention.

However, there is no order. What was done? In which order? At what parameters?

The abstract does not include thesis or purpose, research questions.

I advise you to take a look at the research on Cu-Ni-Si alloys by, for example, Rdzawski, Stobrawa.

Figure 1 caption is illegible. As if taken out of context. Perhaps the drawing is in the wrong place.

The precipitation hardening of Cu-Ni-Si alloys is described in the Materials and Methods chapter, but there is no mention of it in the introduction. Plastic deformation was done cold, but in what condition was the material? After supersaturation or supersaturation and aging?

How were the test alloys made?

How was the chemical composition tested?

How was the preparation for research prepared?

You need to add a table in the methodology that will explain the difference between samples A, b and C. In the table, assign parameters. If they do not differ, what is the purpose of determining samples A, B, C?

Figures 4 and 5 are too small.

I believe that the value of the work would be even more enhanced by analyzing the distribution of elements to show the location of the strengthening phases.

The authors obtained interesting research results, but conclusions need to be changed. That dislocations strengthen it is known. It is also known that the ease of deformation depends on the grain orientation in relation to the stress fields.

Author Response

Answers to the comments by Reviewer #2

We would like to express special thanks for your questions and comments which are reasonable and important. On the all comments, we revised the text and reply the answer. We believe the revised manuscript could meet the all the comments by reviewers.

Comment #1: The work is very interesting and the results of EBSD deserve special attention. However, there is no order. What was done? In which order? At what parameters? The abstract does not include thesis or purpose, research questions. I advise you to take a look at the research on Cu-Ni-Si alloys by, for example, Rdzawski and Stobrawa. Figure 1 caption is illegible. As if taken out of context. Perhaps the drawing is in the wrong place.

Response:

Thank you for pointing out the mistake. Revision has been made in abstract and introduction section. We added the review by Rdzawski in the reference [14].

In Abstract, the revised part is as following,

Before:

This study investigates micro-mechanical properties of a precipitation strengthening-type Cu-Ni-Si alloy (Cu-2.4Ni-0.51Si-9.3Zn-0.15Sn-0.13Mg). Three micro-pillars with a square cross-section of 20 × 20 × 40 μm3 were fabricated by focused ion beam (FIB) micromachining apparatus and tested by a machine specially designed for micro-sized specimens. The three pillars were deformed complicatedly and showed different yield strengths depending on the crystal orientation. The micro-mechanical tests revealed work hardening by the precipitation clearly. Electron backscattered diffraction analysis of a deformed specimen showed a gradual rotation of grain axis at the grain boundaries after the compression test.

After:

The aim of this study is to investigates a characteristic deformation behavior of a precipitation strengthening-type Cu-Ni-Si alloy (Cu-2.4Ni-0.51Si-9.3Zn-0.15Sn-0.13Mg) by micro-compression specimens. Three micro-pillars with a square cross-section of 20 × 20 × 40 μm3 were fabricated by focused ion beam (FIB) micromachining apparatus and tested by a machine specially designed for micro-sized specimens. The three pillars were deformed complicatedly and showed different yield strengths depending on the crystal orientation. The micro-mechanical tests revealed work hardening by the precipitation clearly. Electron backscattered diffraction analysis of a deformed specimen showed a gradual rotation of grain axis at the grain boundaries after the compression test.

In first part of Introduction, we revised as

Before:

When size of materials decreases from bulk size to micrometer scale, the effects of an increase in the ratio of surface area per volume, anisotropy of the grains and the grain boundaries are high. The deformation of metallic materials is fundamentally explained by movement of the dislocations. When the ratio of surface area per volume is high, the dislocations could escape to the free surface before having interaction with each other, and the dislocation density would decrease, which leads to an increase in the apparent strength. This is a widely accepted explanation of size dependent deformation behavior known as the “Size Effect” observed in crystalline metallic materials [1]. The investigation expands to variety of metallic materials such as non-cubic metals [2, 3] and nanocrystalline materials [4, 5]. The dislocation motion could be affected by several internal defects such as vacancies [6], grain boundaries [5] and precipitations [7].

After:

When size of materials decreases from bulk size to micrometer scale, the dislocations could easily escape to the free surface before having interaction with each other, and the dislocation density would decrease, which leads to a dislocation starved condition and cause increased strength. This is a widely accepted explanation of size dependent deformation behavior known as the “Size Effect” observed in crystalline metallic materials [1]. The investigation expands to variety of metallic materials such as non-cubic metals [2, 3] and nanocrystalline materials [4, 5]. The dislocation motion could be affected by several internal defects such as vacancies [6], grain boundaries [5] and precipitations [7].

Also we add the reference as

  • [14] Krupinska, B.; Rdzawski, Z.; Krupinski, M.; Pakieła, W. Precipitation Strengthening of Cu–Ni–Si Alloy. Materials, 2020, 13, 1182

Comment #2: The precipitation hardening of Cu-Ni-Si alloys is described in the Materials and Methods chapter, but there is no mention of it in the introduction. Plastic deformation was done cold, but in what condition was the material? After supersaturation or supersaturation and aging? How were the test alloys made? How was the chemical composition tested? How was the preparation for research prepared?

Response: Thank you for this comment and revised the literature in introduction and 2.1. Sample composition and heat treatment process

In Introduction, we revised as following;

Before:

In this study, micro-mechanical properties of a precipitation hardened copper alloy, Cu-Ni-Si alloy, is investigated. Cu-Ni-Si alloy is a precipitation strengthening-type alloy including d-Ni2Si precipitates and receives a lot of attention for its high strength and high electrical conductivity [12, 13]. On the other hand, effects of the precipitates in the metallic materials by micro-mechanical tests are not thoroughly investigated. In previous studies, we reported micro-tensile tests of a pure copper and a Cu-Ni-Si alloy with micro-tensile specimens fabricated by focused ion beam (FIB) milling system. In the micro-tensile tests, both of the specimens showed characteristic large serrations during deformation, which were not observed in the bulk samples, and the obvious necking deformation led to a decrease in the flow stress [14]. There is still no report on micro-compression of Cu-Ni-Si alloys, and comparisons of the results obtained from compression and tensile tests would be valuable to understand mechanical properties of the alloys in micro-scale. In this study, micro-mechanical property of Cu-2.4Ni-0.51Si-9.3Zn-0.15Sn-0.13Mg alloy by micro-compression test is discussed.

After:

In this study, micro-mechanical properties of a precipitation hardened copper alloy, Cu-Ni-Si alloy, is investigated. Cu-Ni-Si alloy is a precipitation strengthening-type alloy including d-Ni2Si precipitates and receives a lot of attention for its applications for electronic components thanks to their low cost, high strength and good electrical and thermal conductivity [12-14]. These properties are also beneficial for use in micro electro mechanical systems (MEMS). However, effects of the precipitates in micro scales are not yet thoroughly investigated. In our previous studies, we reported micro-tensile tests of a pure copper and a Cu-Ni-Si alloy with micro-tensile specimens fabricated by focused ion beam (FIB) milling system. In the micro-tensile tests, both of the specimens showed characteristic large serrations during deformation, which were not observed in the bulk samples, and the obvious necking deformation led to a decrease in the flow stress [15]. For deep understanding of micro-mechanical behavior in precipitate strengthened metals, comparisons of the results obtained from compression and tensile tests would be valuable to understand mechanical properties of the alloys in micro-scale in order to utilize them to the micro or nano electro mechanical systems. In this study, micro-mechanical property of Cu-2.4Ni-0.51Si-9.3Zn-0.15Sn-0.13Mg alloy by micro-compression test is discussed.

Also in “Materials and Methods”

Before:

In this study, a piece of Cu-Ni-Si alloy composed of Cu-2.4Ni-0.51Si-9.3Zn-0.15Sn-0.13Mg (mass%) was used. For the Cu-Ni-Si alloy, a heat treatment process was conducted first. The heat treatment process involved homogenization at 1223 K for 18 ks, cold rolling for 90% in thickness reduction rate and solution treatment process at 1123 K for 600 seconds. In order to reveal the effect of the precipitation, aging at 723 K for 300 seconds was performed, where the temperature and time needed for the precipitation hardening were determined in previous studies [13, 15]. The grain size of the Cu-Ni-Si alloy was about 30 μm.

After:

The material investigated in this study was supplied by Furukawa Electric Inc. in a sheet form (1.2mm thickness) with the chemical composition Cu-2.4Ni-0.51Si-9.3Zn-0.15Sn-0.13Mg (mass%) . For the Cu-Ni-Si alloy, a heat treatment process was conducted first. The heat treatment process involved homogenization at 1223 K for 18 ks, cold rolling for 90% in thickness reduction rate and solution treatment process at 1123 K for 600 seconds. In order to reveal the effect of the precipitation, aging at 723 K for 300 seconds was performed, where the temperature and time needed for the precipitation hardening were determined in previous studies [13]. The grain size of the Cu-Ni-Si alloy was about 30 μm.

Comment #3: You need to add a table in the methodology that will explain the difference between samples A, b and C. In the table, assign parameters. If they do not differ, what is the purpose of determining samples A, B, C?

Response: Samples A, B and C are identical size and tested in same manner. Only difference is the grain geometry inside pillars as shown in figure 1. The explanation added for clarity.

Before:

Figure 1. SIM images of the three micro-compression specimens (A: (a), (d), B (b), (e), C: (c), (f)) of Cu-Ni-Si alloy by the compression test before: (a), (b), (c) and after; (d), (e), (f).

After:

Figure 1. Scanning ion microscopy images of the three micro-compression specimen AA: (a), (d), specimen B (b), (e),and specimen C: (c), (f)). The image taken before : (a), (b), (c) and after; (d), (e), (f) the compression test.

Comment #4 : Figures 4 and 5 are too small.

Response: The figures are enlarged for clarity.

Comment #5 :I believe that the value of the work would be even more enhanced by analyzing the distribution of elements to show the location of the strengthening phases.

Response : We fully agree to the comment. Showing the location of the strengthening phases and the interaction to dislocations is valuable to investigate micro-mechanical behavior of precipitates. However, TEM observation needed for visualize few nm thick precipitates. Taking the TEM sample from the deformed pillars need considerable additional research activities, which we decided to leave for further investigation.

Comment #6 : The authors obtained interesting research results, but conclusions need to be changed. That dislocations strengthen it is known. It is also known that the ease of deformation depends on the grain orientation in relation to the stress fields.

Response

Thank you for your constructive comment about conclusions. The revision made in conclusion.

Before:

Micro-compression tests of Cu-Ni-Si alloys were conducted. The mechanical tests of three micro-pillars showed different yield strength values ranged from 200 to 300 MPa, and this difference is considered to be contributed by the size effect. All of the engineering stress-engineering strain curves showed work hardening by the precipitations, which is similar to the bulk specimen, and the specimen deformed heterogeneity. In the micro-compression test, the SIM images showed small slip steps, which mean heterogeneous deformation by less restriction between the indenter and the specimen. EBSD analysis indicated a typical gradual rotation of crystal orientation and accumulation of dislocation at the grain boundaries which imply strengthening of grain boundary by the precipitates on the heterogeneous deformation, which came from the grains of the different compressive strength values depending on crystallographic orientation.

After:

Micro-compression tests of Cu-Ni-Si alloys were conducted. The mechanical tests of three micro-pillars showed different yield strength values ranged from 200 to 300 MPa, and this difference is considered to be contributed by the size effect. All of the work hardening curve exhibit plateau regime, which is similar to the bulk specimen, and the specimen deformed heterogeneously. In the micro-compression test, the SIM images showed small slip steps, which mean heterogeneous deformation by less restriction between the indenter and the specimen. EBSD analysis indicated a typical gradual rotation of crystal orientation and high accumulation of dislocation at the grain boundaries which imply enhanced dislocation multiplication at precipitates by Orowan mechanism. The study showed precipitation effectively strengthen the metals and enhance work hardening even in the micro-specimens with few grains inside. The findings indicate the practical applications of precipitation hardened materials for micro-components used in MEMS.

Reviewer 3 Report

The paper is interesting. I have little experience regarding small scale  measurements, so I would have wanted a more basic introduction. The English is sometimes problematic. I am not always sure if the authors choose wrong words. For example in the introduction 30-34:

When the ratio of surface area per volume is high, the dislocations could escape to the free surface before having interaction with each other, and the dislocation density would decrease, which leads to an increase in the apparent strength. This is a widely accepted explanation of size dependent deformation behavior known as the “Size Effect” observed in crystalline metallic materials [1].

The logic is not easy to grasp. Obviously, if the barrier for introducing dislocations is large enough, a dislocation free crystal can be arbitrarily strong. A high density will strengthen the material by dislocations entangled / interacting with each other, as the authors also note further down. A decrease in dislocation density can therefore go both ways regarding strength. This should be explained clearly. 

What do the authors mean by 'apparent strength'? Does it mean it isn't really strength? Explain. Also, they claim that 'this is a widely accepted explanation of size dependent deformation behaviour', and use 1 reference to support this. But this reference is 'just' cited about 120 times since 1961.

48-50:

Cu‐Ni‐Si  alloy  is  a  precipitation  strengthening‐type  alloy  including  δ‐Ni2Si  precipitates and receives a lot of attention for its high strength and high electrical conductivity [10, 11]

The number of references here is on par with the above, but I would love to hear instead why there is attention. Applications?

Fig. 1 looks good, but explanation is lacking. I would like to know the crystallographic directions   in a->c. (e.g. is it the <100>Cu direction pointing upwards?) Also, indicate the direction(s) of the applied force . It would make this paper more readable for others than experts. The caption misses  ':' after 'B'. Point out what are twin and grain boundaries, and explain the contrasts.

65: 'a piece of' is not good enough. When doing heat-treatments to obtain precipitation, sample size is crucial. Precipitation is also 'size dependent'. The temperatures of the performed heat-treatments are high here. I actually wonder if the specimens contain precipitates at all. The smaller the piece, the relatively more solutes and vacancies leave to the surfaces. For example, in alloys heat-treated in TEM, (10-100 nm thickness) using a heating holder and normal ageing procedures (which should cause a lot of precipitation), often no precipitation at all is seen. Be aware that most internal surfaces are connected. Dislocations run between grains, and all grain-boundaries connect and lead to the outer surfaces. Thus, one may be doing experiments on a completely different alloy than intended. The authors should verify and discuss this. 

121: The  work  hardening  was  caused  by  obstruction  of  dislocation  by  the  precipitates. Was it?  At least it breaks with my (and I believe, conventional) understanding of work-hardening. Isn't work-hardening a function of the dislocation density? The more you work the material, the more dislocations you  produce, the harder slip /deformation becomes, the stronger the material? 

135-141: Here the confusion is repeated. Maybe it is language problems, but the authors seem to mix hardening by precipitates and dislocations.  

Fig. 4 and the text 151-162: It is refered to 'directions' in the figure, but none are indicated. Shear-lines are not indicated.

Fig. 5. Few are familiar with pole figures. A few sentences would be appropriate to explain how the assemble of dark spots in the corners of the figures, as well as the colors, relate to the projection in (b). Is it obvious that (b) is a view of (a) along the axis? Why not try to make it easier for the reader, some of which may be students trying to learn? 

In the conclusion is said: All of the engineering stress‐engineering strain curves  showed  work  hardening  by  the  precipitations,  which  is  similar  to  the  bulk  specimen,  and  the  specimen deformed heterogeneity

So it seem it is really a confusion about precipitation hardening and work-hardening, which is the main reason for rejecting the paper.

Author Response

Answers to the comments by Reviewer #3

We would like to express special thanks for your questions and comments which are reasonable and important. On the all comments, we revised the text and reply the answer. We believe the revised manuscript could meet the all the comments by reviewers.

Comment #1:The paper is interesting. I have little experience regarding small scale  measurements, so I would have wanted a more basic introduction. The English is sometimes problematic. I am not always sure if the authors choose wrong words. For example in the introduction 30-34:

When the ratio of surface area per volume is high, the dislocations could escape to the free surface before having interaction with each other, and the dislocation density would decrease, which leads to an increase in the apparent strength. This is a widely accepted explanation of size dependent deformation behavior known as the “Size Effect” observed in crystalline metallic materials [1].

The logic is not easy to grasp. Obviously, if the barrier for introducing dislocations is large enough, a dislocation free crystal can be arbitrarily strong. A high density will strengthen the material by dislocations entangled / interacting with each other, as the authors also note further down. A decrease in dislocation density can therefore go both ways regarding strength. This should be explained clearly. 

What do the authors mean by 'apparent strength'? Does it mean it isn't really strength? Explain. Also, they claim that 'this is a widely accepted explanation of size dependent deformation behaviour', and use 1 reference to support this. But this reference is 'just' cited about 120 times since 1961.

48-50:Cu‐Ni‐Si  alloy  is  a  precipitation  strengthening‐type  alloy  including  δ‐Ni2Si  precipitates and receives a lot of attention for its high strength and high electrical conductivity [10, 11]

The number of references here is on par with the above, but I would love to hear instead why there is attention. Applications?

Response

Thank you for your comment regarding the introduction. The introduction section modified accordingly.

In the introduction 30-34:

Before:

When size of materials decreases from bulk size to micrometer scale, the effects of an increase in the ratio of surface area per volume, anisotropy of the grains and the grain boundaries are high. The deformation of metallic materials is fundamentally explained by movement of the dislocations. When the ratio of surface area per volume is high, the dislocations could escape to the free surface before having interaction with each other, and the dislocation density would decrease, which leads to an increase in the apparent strength. This is a widely accepted explanation of size dependent deformation behavior known as the Size Effectobserved in crystalline metallic materials [1]. The investigation expands to variety of metallic materials such as non-cubic metals [2, 3] and nanocrystalline materials [4, 5]. The dislocation motion could be affected by several internal defects such as vacancies [6], grain boundaries [5] and precipitations [7].

After:

When size of materials decreases from bulk size to micrometer scale, the dislocations could easily escape to the free surface before having interaction with each other, and the dislocation density would decrease, which leads to a dislocation starved condition and cause increased strength. This is a widely accepted explanation of size dependent deformation behavior known as the “Size Effect” observed in crystalline metallic materials [1]. The investigation expands to variety of metallic materials such as non-cubic metals [2, 3] and nanocrystalline materials [4, 5]. The dislocation motion could be affected by several internal defects such as vacancies [6], grain boundaries [5] and precipitations [7].

In lines 48-50, we revised as following;

Before:

In this study, micro-mechanical properties of a precipitation hardened copper alloy, Cu-Ni-Si alloy, is investigated. Cu-Ni-Si alloy is a precipitation strengthening-type alloy including d-Ni2Si precipitates and receives a lot of attention for its high strength and high electrical conductivity [12, 13]. On the other hand, effects of the precipitates in the metallic materials by micro-mechanical tests are not thoroughly investigated. In previous studies, we reported micro-tensile tests of a pure copper and a Cu-Ni-Si alloy with micro-tensile specimens fabricated by focused ion beam (FIB) milling system. In the micro-tensile tests, both of the specimens showed characteristic large serrations during deformation, which were not observed in the bulk samples, and the obvious necking deformation led to a decrease in the flow stress [14]. There is still no report on micro-compression of Cu-Ni-Si alloys, and comparisons of the results obtained from compression and tensile tests would be valuable to understand mechanical properties of the alloys in micro-scale. In this study, micro-mechanical property of Cu-2.4Ni-0.51Si-9.3Zn-0.15Sn-0.13Mg alloy by micro-compression test is discussed.

After:

In this study, micro-mechanical properties of a precipitation hardened copper alloy, Cu-Ni-Si alloy, is investigated. Cu-Ni-Si alloy is a precipitation strengthening-type alloy including d-Ni2Si precipitates and receives a lot of attention for its applications for electronic components thanks to their low cost, high strength and good electrical and thermal conductivity [12-14]. These properties are also beneficial for use in micro electro mechanical systems (MEMS). However, effects of the precipitates in micro scales are not yet thoroughly investigated. In our previous studies, we reported micro-tensile tests of a pure copper and a Cu-Ni-Si alloy with micro-tensile specimens fabricated by focused ion beam (FIB) milling system. In the micro-tensile tests, both of the specimens showed characteristic large serrations during deformation, which were not observed in the bulk samples, and the obvious necking deformation led to a decrease in the flow stress [15]. For deep understanding of micro-mechanical behavior in precipitate strengthened metals, comparisons of the results obtained from compression and tensile tests would be valuable to understand mechanical properties of the alloys in micro-scale in order to utilize them to the micro or nano electro mechanical systems. In this study, micro-mechanical property of Cu-2.4Ni-0.51Si-9.3Zn-0.15Sn-0.13Mg alloy by micro-compression test is discussed.

Comment#2: Fig. 1 looks good, but explanation is lacking. I would like to know the crystallographic directions   in a->c. (e.g. is it the <100>Cu direction pointing upwards?) Also, indicate the direction(s) of the applied force . It would make this paper more readable for others than experts. The caption misses  ':' after 'B'. Point out what are twin and grain boundaries, and explain the contrasts.

Response: We revised the caption of Figure 1,

Before:

Figure 1. SIM images of the three micro-compression specimens (A: (a), (d), B (b), (e), C: (c), (f)) of Cu-Ni-Si alloy by the compression test before: (a), (b), (c) and after; (d), (e), (f).

After:

Figure 1. Scanning ion microscopy images of the three micro-compression specimen AA: (a), (d), specimen B (b), (e),and specimen C: (c), (f)). The image taken before : (a), (b), (c) and after; (d), (e), (f) the compression test.

Comment#3: 65: 'a piece of' is not good enough. When doing heat-treatments to obtain precipitation, sample size is crucial. Precipitation is also 'size dependent'. The temperatures of the performed heat-treatments are high here. I actually wonder if the specimens contain precipitates at all. The smaller the piece, the relatively more solutes and vacancies leave to the surfaces. For example, in alloys heat-treated in TEM, (10-100 nm thickness) using a heating holder and normal ageing procedures (which should cause a lot of precipitation), often no precipitation at all is seen. Be aware that most internal surfaces are connected. Dislocations run between grains, and all grain-boundaries connect and lead to the outer surfaces. Thus, one may be doing experiments on a completely different alloy than intended. The authors should verify and discuss this. 

Response

Thank you for pointing out the mistakes, the descriptions are revised.

In 2.1. Sample composition and heat treatment process

Before:

In this study, a piece of Cu-Ni-Si alloy composed of Cu-2.4Ni-0.51Si-9.3Zn-0.15Sn-0.13Mg (mass%) was used. For the Cu-Ni-Si alloy, a heat treatment process was conducted first. The heat treatment process involved homogenization at 1223 K for 18 ks, cold rolling for 90% in thickness reduction rate and solution treatment process at 1123 K for 600 seconds. In order to reveal the effect of the precipitation, aging at 723 K for 300 seconds was performed, where the temperature and time needed for the precipitation hardening were determined in previous studies [13, 15]. The grain size of the Cu-Ni-Si alloy was about 30 μm.

After:

The material investigated in this study was supplied by Furukawa Electric Inc. in a sheet form (1.2mm thickness) with the chemical composition Cu-2.4Ni-0.51Si-9.3Zn-0.15Sn-0.13Mg (mass%) . For the Cu-Ni-Si alloy, a heat treatment process was conducted first. The heat treatment process involved homogenization at 1223 K for 18 ks, cold rolling for 90% in thickness reduction rate and solution treatment process at 1123 K for 600 seconds. In order to reveal the effect of the precipitation, aging at 723 K for 300 seconds was performed, where the temperature and time needed for the precipitation hardening were determined in previous studies [13]. The grain size of the Cu-Ni-Si alloy was about 30 μm.

Also in 2.2. Fabrication process

Before

The micro-sized specimens were fabricated by a FIB (Hitachi FB2100). For the compression test, three micro-pillars (specimens A, B and C) as shown in Figure 1, having size of 20 x 20 μm2 in cross-section and 40 μm in length were fabricated. For fabrication of the pillar, the specimen was tilted at 45±2 degrees to avoid tapering of the pillar [15, 16].

After

The micro-sized specimens were fabricated by a FIB (Hitachi FB2100) from the bulk sample described in former section 2.1 [16, 17]. In the beginning, the sliced sample with the thickness of about 50mm was fabricated from middle part of the bulk sample. For fabrication of the pillar, the specimen was tilted at 45±2 degrees to avoid tapering of the pillar. For the compression test, three micro-pillars (specimens A, B and C) with different grain geometry as shown in Figure 1, having identical size of 20 x 20 μm2 in cross-section and 40 μm in length were fabricated.

Comment#4: 121: The  work  hardening  was  caused  by  obstruction  of  dislocation  by  the  precipitates. Was it?  At least it breaks with my (and I believe, conventional) understanding of work-hardening. Isn't work-hardening a function of the dislocation density? The more you work the material, the more dislocations you produce, the harder slip /deformation becomes, the stronger the material? 

135-141: Here the confusion is repeated. Maybe it is language problems, but the authors seem to mix hardening by precipitates and dislocations.  

Response

Thank you for pointing out the mistakes, the descriptions are revised.

In line 135-141:

Before:

Figure 3 shows work hardening behavior of specimen A in the micro-compression test and indicates the enhanced work hardening. The work hardening rate; Q was calculated by the following equation

Q=Ds/De    (1)

using the true stress; e and true strain; s which were reduced from the engineering stress and the engneering strain observed, respectively. This behavior shows obstruction of the dislocation movement by the precipitations.

After:

Figure 3 shows work hardening behavior of specimen A in the micro-compression test and indicates the enhanced work hardening. The work hardening rate; Q was calculated by the following equation

Q=Ds/De    (1)

using the true stress; e and true strain; s which were reduced from the engineering stress and the engneering strain observed, respectively. In the compression test, repetition of large bursts and drops could not be observed unlike the micro-tensile test reported in a previous study [15]. The steady state of work hardening rate curve observed at about 600 MPa may have been caused by the precipitation.

Comment#5: Fig. 4 and the text 151-162: It is referred to 'directions' in the figure, but none are indicated. Shear-lines are not indicated.

Response: Thank you for comments, the descriptions are revised.

In text 151-162, we revised as following;

Before:

Figure 4. SIM images of sample A in Cu-Ni-Si alloy specimens after micro-compression test, observed from three different directions.

After:

Figure 4. SIM images of sample A in Cu-Ni-Si alloy specimens after micro-compression test, observed taken from the front (a) and sides (b),(c).

Comment#6: Fig. 5. Few are familiar with pole figures. A few sentences would be appropriate to explain how the assemble of dark spots in the corners of the figures, as well as the colors, relate to the projection in (b). Is it obvious that (b) is a view of (a) along the axis? Why not try to make it easier for the reader, some of which may be students trying to learn? 

Response:

Thank you for your comment. Manuscript has been revised accordingly.

In text 162-, we revised as following

Before:

Figure 5 shows crystal characteristic of the Cu-Ni-Si alloy specimen A after the compression test. The upper square plane has two grains of a and b. Crystallographic analysis indicates that grain a and b have near <110> and <111> orientations at surface, respectively.

After:

Figure 5 shows crystal characteristic of the Cu-Ni-Si alloy specimen A after the compression test. The color in figure 5b represents the orientations in inverse pole figure (IPF) against pillar axis shown in the inset. The orientations of grains a and b are plotted in IPF shown in figure 5c and 5d. Grain a and b had near <110> and <111> orientations at surface, respectively.

Comment#7: In the conclusion is said: All of the engineering stress‐engineering strain curves showed  work  hardening  by  the  precipitations,  which  is  similar  to  the  bulk  specimen,  and  the specimen deformed heterogeneity. So it seem it is really a confusion about precipitation hardening and work-hardening, which is the main reason for rejecting the paper.

Response

Thank you for your constructive comment about conclusions. The revision made in conclusion.

Before:

Micro-compression tests of Cu-Ni-Si alloys were conducted. The mechanical tests of three micro-pillars showed different yield strength values ranged from 200 to 300 MPa, and this difference is considered to be contributed by the size effect. All of the engineering stress-engineering strain curves showed work hardening by the precipitations, which is similar to the bulk specimen, and the specimen deformed heterogeneity. In the micro-compression test, the SIM images showed small slip steps, which mean heterogeneous deformation by less restriction between the indenter and the specimen. EBSD analysis indicated a typical gradual rotation of crystal orientation and accumulation of dislocation at the grain boundaries which imply strengthening of grain boundary by the precipitates on the heterogeneous deformation, which came from the grains of the different compressive strength values depending on crystallographic orientation.

After:

Micro-compression tests of Cu-Ni-Si alloys were conducted. The mechanical tests of three micro-pillars showed different yield strength values ranged from 200 to 300 MPa, and this difference is considered to be contributed by the size effect. All of the work hardening curve exhibit plateau regime, which is similar to the bulk specimen, and the specimen deformed heterogeneously. In the micro-compression test, the SIM images showed small slip steps, which mean heterogeneous deformation by less restriction between the indenter and the specimen. EBSD analysis indicated a typical gradual rotation of crystal orientation and high accumulation of dislocation at the grain boundaries which imply enhanced dislocation multiplication at precipitates by Orowan mechanism. The study showed precipitation effectively strengthen the metals and enhance work hardening even in the micro-specimens with few grains inside. The findings indicate the practical applications of precipitation hardened materials for micro-components used in MEMS.

Round 2

Reviewer 2 Report

No more comments. 

greetings

Reviewer 3 Report

The paper is now fine for publication. The authors did an effort to clear up the misunderstandings that existed in the first version. The reformulations cleared up the misunderstandings